# Impact of chart image characteristics on stock price prediction with a convolutional neural network

**Guangxun Jin**[1], **Ohbyung Kwon**[2]*

**1** Department of Management, Kyung Hee University, Seoul, South Korea, **2** School of Management, Kyung Hee University, Seoul, South Korea

* obkwon@khu.ac.kr

## Abstract

Stock price prediction has long been the subject of research because of the importance of accuracy of prediction and the difficulty in forecasting. Traditionally, forecasting has involved linear models such as AR and MR or nonlinear models such as ANNs using standardized numerical data such as corporate financial data and stock price data. Due to the difficulty of securing a sufficient variety of data, researchers have recently begun using convolutional neural networks (CNNs) with stock price graph images only. However, we know little about which characteristics of stock charts affect the accuracy of predictions and to what extent. The purpose of this study is to analyze the effects of stock chart characteristics on stock price prediction via CNNs. To this end, we define the image characteristics of stock charts and identify significant differences in prediction performance for each characteristic. The results reveal that the accuracy of prediction is improved by utilizing solid lines, color, and a single image without axis marks. Based on these findings, we describe the implications of making predictions only with images, which are unstructured data, without using large amounts of standardized data. Finally, we identify issues for future research.

⊝ OPEN ACCESS

**Data Availability Statement:** The data (data set and meta data) underlying the results presented in the study are available from: [1] Variable description and experimental data Link https://figshare.com/articles/dataset/Variable_description_

## I. Introduction

Successful stock trading is highly important for investors. Considering multiple stocks when trading, they must buy or sell by selecting appropriate stocks with attention to the timing of the sale. Accordingly, stock price prediction is a long-standing research issue. Because stock prices are determined by a wide variety of variables [1], prediction seems to be a random walk, especially using past information [2].

Stock price prediction has traditionally been performed using linear models such as AR, ARMA, and ARIMA and its variations [3–5]. However, the assumption that stock price fluctuations are linear is an oversimplification of the factors affecting stock prices; other assumptions are that hidden dynamics are at work and only overall trends can be seen [6,7]. Nonlinear models such as artificial neural networks (ANNs) and convolutional neural networks (CNNs) can also be used to predict stock prices [6,8]. Deep neural networks (DNNs), which utilize

and_experimental_data/14074496 [2] Company-dataset Link: https://figshare.com/articles/dataset/dataset/14074502 [3] Image-dataset Link: https://figshare.com/articles/dataset/cnn-dataset/14074292 The authors confirm that the authors of the present study had no special privileges in accessing these datasets which other interested researchers would not have.

**Funding:** This work was supported by the Ministry of Education of the Republic of Korea and the National Research Foundation of Korea (NRF-2020S1A5B8103855).

**Competing interests:** No authors have competing interests.

deep learning algorithms for stock price prediction, may also been used. Although recurrent neural networks (RNNs) and the like are also being proposed [9], CNNs appear to be relatively superior for the purpose of stock price prediction [10].

Existing studies of stock price prediction mainly use numerical data. In recent years, attempts have begun to predict stock prices through image-based discrimination using the stock price graph itself as input data without using profile information or numerical data [2,11,12]. Even stock forecasting experts use the stock chart itself to make predictions. The reason for this is that the chart contains information that can form the basis for prediction; the extracted knowledge pertains highly to stock price prediction. In this study, we examine the possibility that this knowledge can be extracted using a deep learning model. Some previous studies involving deep learning-based stock price prediction have used additional charts as training data. However, most stock price chart types are fixed so that the characteristics of the chart image that can affect stock price prediction are not easily understood [2,12]. In deep learning algorithms such as CNNs, since the characteristics of the image affect predictive performance, the shape of the chart may be an important variable [11,13]. However, only a few studies examine how the characteristics of stock price charts affect the performance of deep learning algorithms (especially CNNs) for stock price prediction.

The purpose of this study is to analyze the effects of stock price chart characteristics on the stock price prediction performance of CNNs [14]. The present study focuses on improving the quality of stock price image data by implementing a rigorous preprocessing technique: selecting optimal image characteristics. To this end, various types of images were generated from actual stock price data and significant differences in CNN prediction performance were identified for each characteristic. CNN parameters included in this study are dropout, number of filters, and activation functions, as these can strongly affect inference performance. The results of this study reveal that stock prices can be predicted with images only using CNNs.

The findings of this experimental paper are as follows. First, even if no numerical information is prepared, stock price predictions can be made with a high level of accuracy using only image data. Second, when predicting using image data, the preprocessing step of selecting an optimal image shape that can increase accuracy is absolutely necessary. Our experiments identify which factors affect prediction accuracy and which characteristics of charts are useful to increase that accuracy. We hope that our results may be useful for deep learning practitioners to select optimal hyperparameters in a minimal amount of time for the purpose of stock price prediction [15].

## II. Background

### 2.1 Stock price prediction

As previously mentioned, stock price prediction has traditionally been performed using linear models such as AR, ARMA, and ARIMA and its variations [3–5]. In these models, the stock price is used as in a time series model as an independent variable for forecasting. For multivariate models, the company profile (returns, sales, various financial ratios, etc.) [16], stock trading volume [17], stock price trends over a certain period of time, and the number of times a specific company's name was mentioned in search engines [18] have been proposed (see Table 1).

Recently, attempts have been made to predict stock prices using nonlinear models. Various classification algorithms such as ANNs, naïve Bayes, support vector machines (SVMs), and random decision forests have been used. Nonlinear classifiers for stock price prediction show better performance than existing linear models [3–5,10]. Among various options, artificial neural networks (ANNs) have provided good predictive performance and become widely used

**Table 1. Literature on stock price prediction.**

| Method | Determinants | Data Frequency | References |
|---|---|---|---|
| SVM | Single words, bigram, polarity, noun phrase | Daily | [19] |
| ANN, SVM, linear regression | Economic info, financial ratios | Yearly | [16] |
| MLP, RNN, CNN | Close price | Daily | [10] |
| ANN, genetic algorithm | Close price | Daily | [20] |
| LSTM | Rate of return | Daily | [21] |
| CNN | Open and close prices | Daily | [7] |
| CNN | Close price | Monthly | [12] |
| ARIMA, LSTM, CNN | DJIA index closed | Daily | [22] |
| SVM, CNN | Close price | Every 30 minutes | [2] |
| LSTM, CNN | Close price | Daily | [23] |
| LSTM, CNN | Four commodity futures and two equity index closed | Every 5 minutes | [24] |

financial forecasting tools [25–27]. For example, a comparison between ANN and SVM classifiers revealed that ANNs perform better than SVMs in forecasting on the Istanbul Stock Exchange [25]. In order to improve the predictive performance of ANNs, certain features need to be more easily extracted; therefore, feature engineering methods such as principal component analysis (PCA) are now being used to improve the performance of ANNs [28]. However, ANNs do not always perform well. In another study, ten technical indicators were used to classify the up and down movements of stocks using ANN, SVM, random forest, and naïve Bayes classifiers. The authors found that the random forest model outperformed the others [29]. Therefore, a hybrid method has been proposed that combines several prediction techniques such as finding the initial weight of the ANN using a genetic algorithm and a simulated annealing algorithm and then learning the network using a back propagation algorithm. This combinative approach had better results than the standard ANN method [20].

Recently, research was conducted to predict stock prices using deep learning algorithms. For example, DNNs and numerical data have been used [30,31] and long short-term memory (LSTM) or RNN classifiers have been used with unstructured data [32,33]. In general, deep learning-based stock price prediction outperforms conventional stock price prediction in the sense that deep neural network architectures are capable of capturing hidden dynamics and making more accurate predictions [6]. Inference is also possible with a large number of input features and a convolution technique unique to deep learning [7]. However, when applying a nonlinear model such as those involving deep learning algorithms, there is a concern about overfitting. Among deep learning algorithms for stock price prediction, DNNs and RNNs have been proposed in addition to CNNs [9], but CNNs appear to be relatively superior in terms of performance [10]. However, this method has the disadvantage that a large amount of data must be prepared, which requires considerable cost and time. Data availability and data preparation costs are also important when corporate information is not well shared. Despite these difficulties, the input data and processing methods involved in CNNs play an important role in identifying the quality of the extracted features and predicting stock prices. For example, when performing stock price prediction with CNNs using 10-day data from 100 companies listed in Borsa Istanbul, the prediction performance improved to some extent by combining similar features, generating technical indicators, and identifying time-lagged features [34].

While most existing studies of stock price prediction as described above were based on structural data, big data research has recently progressed to stock price prediction using informal data such as text and images. In particular, information from social media on corporate activities is unstructured text information. Social media provides information that can be

determined as favorable or unfavorable to companies through sentiment analysis [35]. How-ever, it is difficult to explain the rise and fall of stock prices using only data based on sentiment analysis. For this reason, big data researchers combine this data with other variables to make predictions, but even in this situation, the problem of data availability reappears.

## 2.2 Stock price prediction based on image-based deep learning

In recent years, attempts to predict stock prices through image-based discrimination have begun using the stock price graph itself as input data without using profile information or numerical data related to stock prices [2,11,12]. This is due to the assumption that future stock price predictions can be made more accurately by learning trends of the past and the premise that a pattern similar to those in the past will occur repeatedly [12]. When deep learning algorithms such as CNNs are used for stock price prediction, they predict the stock price only, not using an image as a dataset [6,8]. The fact that stock prediction analysis experts are also making predictions based on the image characteristics shown in stock charts implies that the chart image contains information that can aid prediction. Image characteristics can provide clues; the knowledge these images provide can be extracted using deep learning models such as CNNs.

## III. Methods

This study examines the effects of the features of images from various stock charts on the accu-racy of stock price predictions, focusing in particular on the role of filter and dropout, the main hyperparameters of CNNs, as mediating variables between image features and prediction performance (see Fig 1). To verify this role, we propose the following hypotheses:

[Hypothesis 1] Image characteristics of stock charts affect the accuracy of stock price pre-diction by CNNs.

[Hypothesis 2] The filter parameter plays a mediating role between image composition vari-ables and prediction accuracy.

[Hypothesis 3] The dropout parameter plays a mediating role between image composition variables and prediction accuracy.

## 3.1 Image characteristics

Table 2 shows the image characteristics of the stock price chart to be considered in this study. Several types of charts were targeted: plots, barplots, and histograms, which are the most com-monly used stock price charts. Second, we determined whether the direction of the chart (hori-zontal or vertical) also affects discrimination and prediction performance. Third, an axis heading displayed can also affect discrimination and prediction performance. Explanations or scale marks on the x- or y-axis provide helpful information for understanding the chart for human analysts, but in the case of CNNs, if text information is not interpreted and is viewed only as part of the image, it may be considered noise. Therefore we included it in the analysis. Fourth, we compare performances between two different forecasting dates: the forecast for the next day of the period shown in the stock price chart, or the forecast 5 days later (i.e., one week later, excluding the weekend). In the literature, there are studies that examine predictions after 1 day and studies that do so after 5 days [2,12]. Fifth, whether data on the chart is displayed in the form of a bar or a line has a similar meaning to the presence or absence of color on the sur-face (see below). If it is displayed as a bar, the thickness and shape of the bar are reflected in learning, and if it is expressed as a line rather than as a bar, it corresponds to the narrowing technique in image processing, meaning that only direction and length information is pro-vided. Sixth, whether the bars and lines are solid or dotted can affect performance.

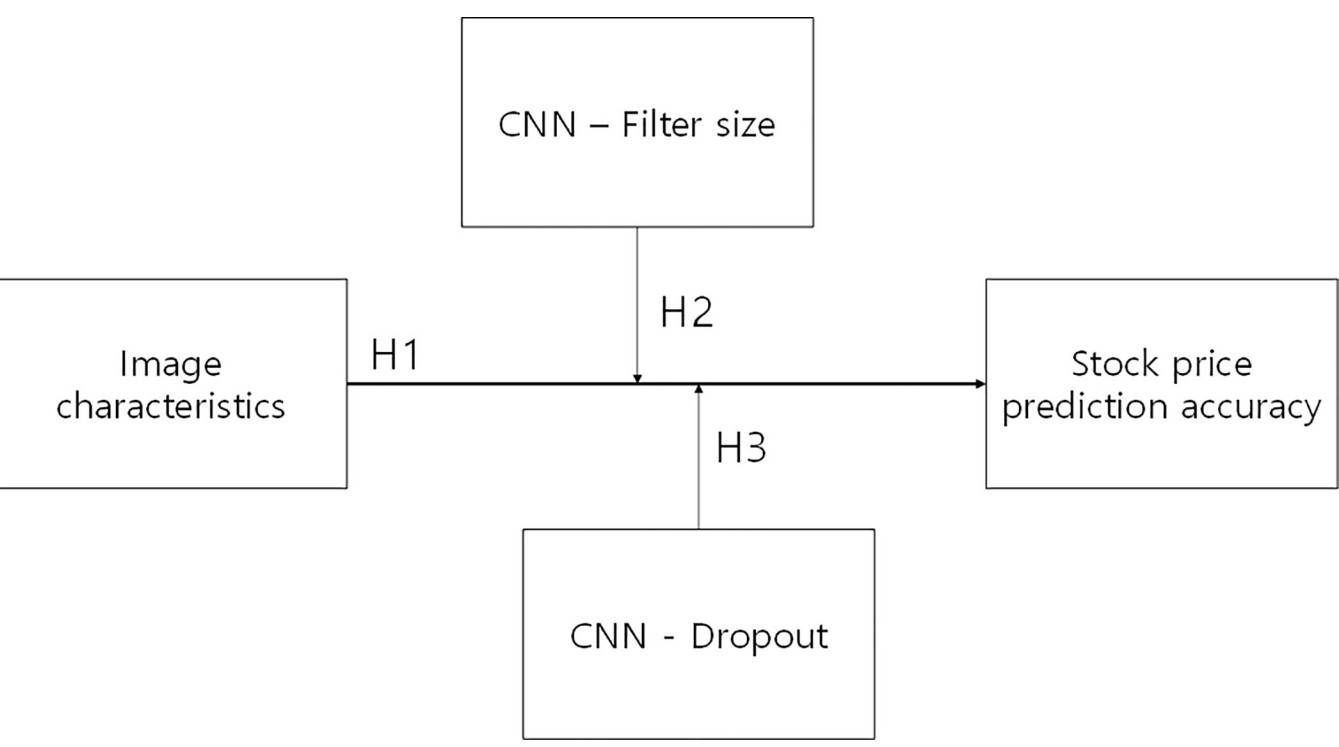

**Fig 1. Research model.**

Finally, the presence or absence of color on the surface of the chart can also affect discrimination and performance. If there is color on the surface, it may be recognized as an area, and if not, it may be recognized as an outline. In image processing, there is a skeletonization process [36]. Combining the skeletonized algorithm with the CNN as the recognition algorithm reduces the impact of the shooting angle and environment on the recognition effect and improves the accuracy of gesture recognition in complex environments [37]. In the context of stock price charts, a color on the side provides information indicating the share of the stock price in the overall image, and no color means it is seen as a line, which means that the degree of change in the stock price is emphasized.

Also, depending on the problem area to be identified, the presence or absence of color may be advantageous for discrimination. Even black, white, and gray color may be advantageous, especially in fields such as medical imaging diagnosis [38] or security search [39]. In some cases, preprocessing, or graying, in a color image is performed in order to remove unnecessary

**Table 2. Image characteristics.**

| Image characteristics | Variable name | Values |
|---|---|---|
| Series | X1 | time series = 1, frequency = 2 |
| Graph type | X2 | plot = 1, hist = 2, barplot = 3, stripchart = 4 |
| Graph direction | X3 | horizontal = 1, vertical = 2 |
| Appearance of axis | X4 | no = 0, yes = 1 |
| Forecast date | X5 | after 1 day = 1, after 5 days = 2 |
| Bar/line | X6 | bar = 1, line = 2 |
| Color | X7 | no = 0, yes = 1 |
| Color tone | X8 | monotone = 0, colorful = 1 |
| Dotted/solid line | X9 | solid line = 0, dotted = 1 (density) |

features that interfere with interpretation. Thus, both black and white and color images affect discrimination and prediction performance. This is especially true when it is difficult to give a special meaning to a color.

### 3.2 CNN characteristics

CNNs are feed forward neural networks that have a unique effect on graphic image processing. A convolutional layer and a pooling layer are included in the network structure. CNNs are particularly widely used as deep learning networks in many recent studies. Representative CNNs include LeNet-5, VGG, and AlexNet. In a fully connected neural network, all neurons in adjacent layers are connected together and the number of outputs can be arbitrarily determined. However, the problem with fully connected neural networks is that the form of data is not represented. For example, input data in the form of an image is composed of the height, length, and channel of the three-dimensional shape (H, W, C). However, the input data, which is a fully connected neural network, must be compressed from 3D data to 1D data. With a MNIST data set, for example, the data image must be input in 784 (28*28) format with 3D data consisting of 1 channel, 28 pixels high and 28 pixels long, or (28,28,1).

Since image information is compressed into one dimension, some spatial information is lost during this process. On the other hand, since CNNs input 3D data and the output value is also 3D, they are more appropriate for classifying images, as the form can be maintained and the data type need not be changed. Therefore, in this paper, we selected a CNN algorithm for the prediction of stock prices using graphic data (see Fig 2).

In addition to the image characteristics considered after image formation, the shape of the dropout and filter can also affect CNN performance. In fact, there are thousands of network characteristics that affect CNNs. Therefore, in this study, only the numbers of dropouts and filters were considered (see Table 3).

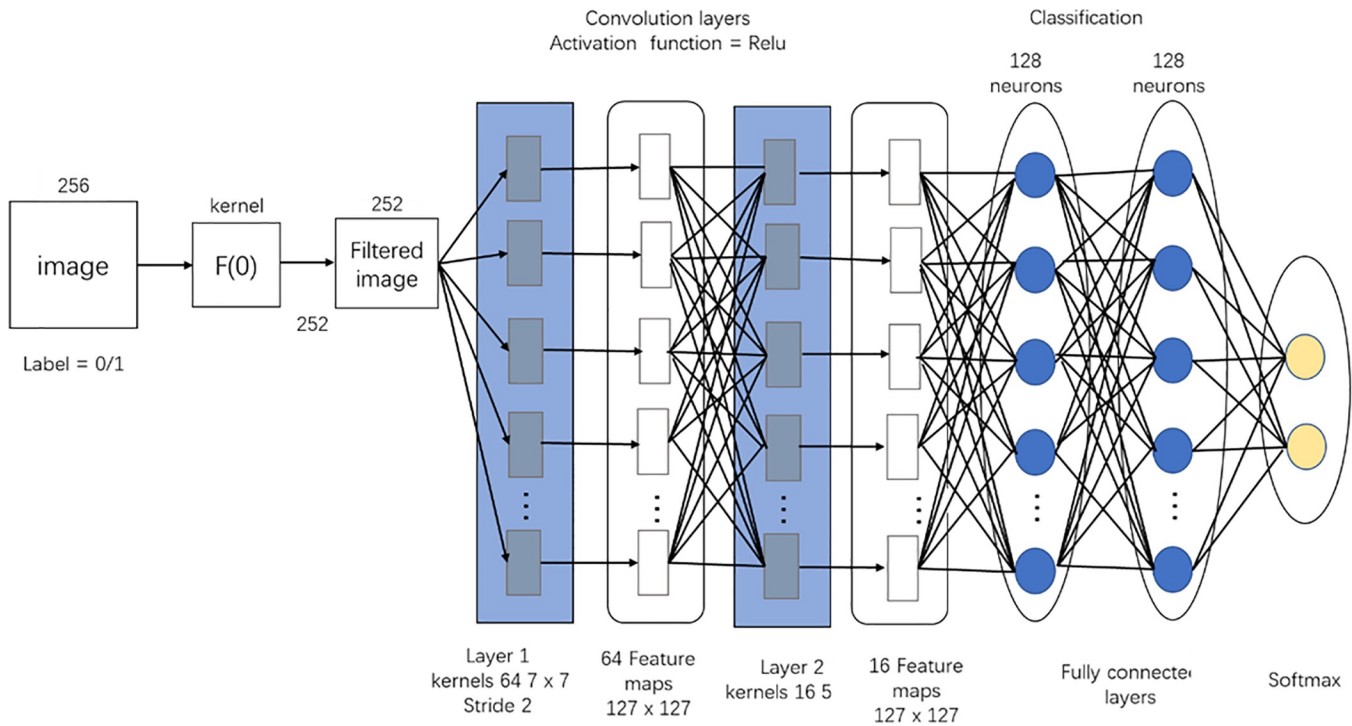

**Fig 2. CNN network.**

Table 3. CNN characteristics.

| Hyperparameter | Values |
|---|---|
| dropout | (0.25, 0.25), (0.5,0.5) |
| filter | (2,4,8),(3,6,12),(4,8,16),(5,10,20),(6,12,24),(7,14,28),(8,16,32),(9,18,36)(10,20,40),(11,22,44), (12,24,48),(13,26,52),(14,28,56),(15,30,60),(16,32,64) |

## IV. Experiment

### 4.1 Data

To create a stock price chart, we first collected daily closing prices for the 5 years from 2015 to 2019 for all 789 companies listed in the KOSPI Index from Dataguide. Note that companies closed during this period were excluded, and non-business days were excluded for other companies. Next, a chart image was created for each company at the closing price for each period of 30 days, with attention to parameters such as vertical/horizontal, line/bar, and colored/colorless for three types of charts: barplot, plot, and histogram. In total, 30 types of chart images were created by making various changes in chart characteristics. Examples of the generated chart images are shown in Fig 3.

In total, 45,407 images were created for each chart, and 1,424,065 images were collected to make up the image dataset. Charts were created by collecting data for each company every 30 days throughout the 5-year period. Labels were automatically generated for each image, and each image was compared to the closing price of stocks 5 days later. Fig 4 provides an example.

Changes in the closing price on the forecast date reveal how much the price rose or fell (after 1 day or 5 days) based on the last day of the 30 days after the image was created (see Eqs (1), (2)). If the image name rate of return was greater than 0, 1 was automatically added to the

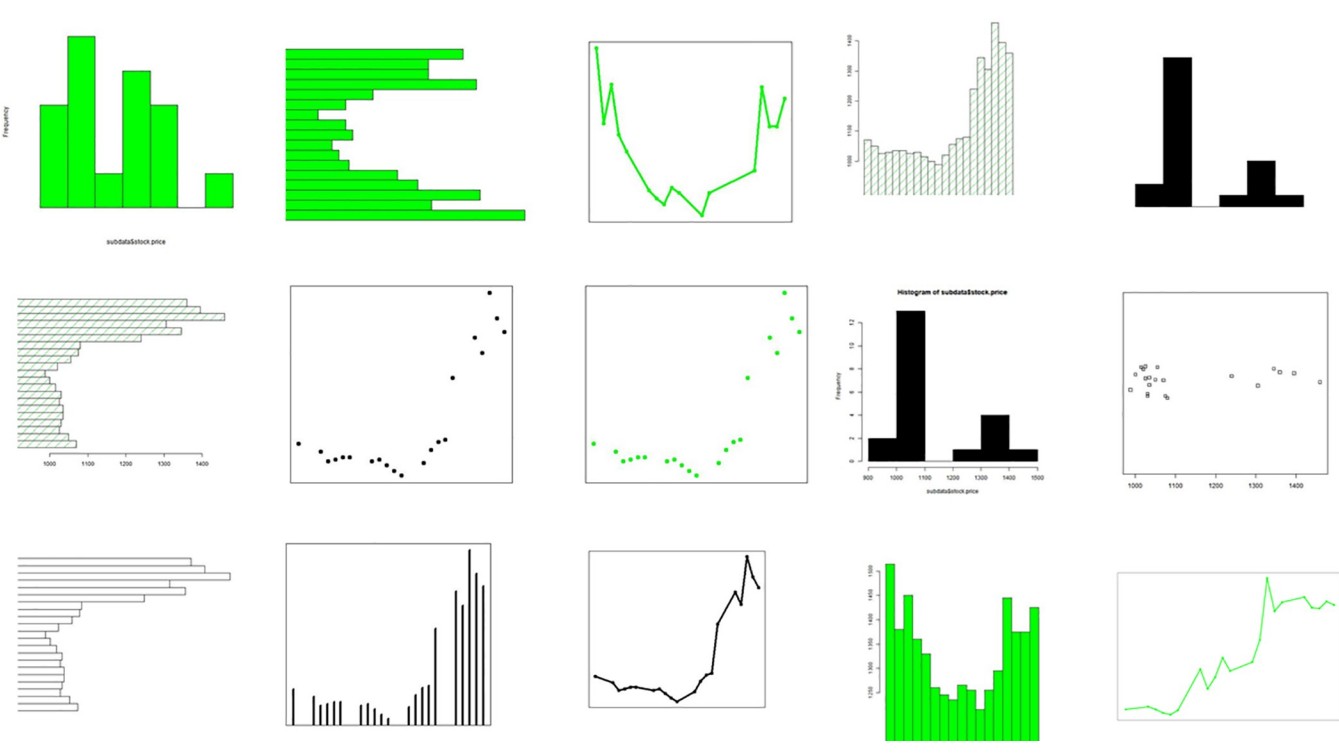

Fig 3. Sample charts.

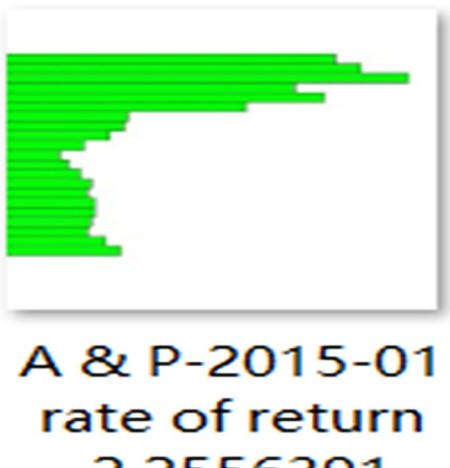

**Fig 4. Sample chart image.**

end of the previously created label; otherwise, 0 was automatically added. If the result was greater than 0, the value was added to the stock price increase dataset; if it was less than or equal to 0, the value was added to the stock price decline dataset. Training and verification data were randomly arranged in a ratio of 9:1.

$$\text{Yield after 5 days} = \frac{value35 - value30}{value30} * 100 \qquad (1)$$

$$\text{Yield after 1 day} = \frac{value31 - value30}{value30} * 100 \qquad (2)$$

## 4.2 CNN model

In this paper, the CNN model structure was constructed using Keras. Keras is an advanced ANN API that enables stock price prediction with the simplest possible code [40]. Keras can also shift computing from CPU to GPU acceleration without code changes.

In the learning phase, a CNN was constructed to train and test the above-described data. The constructed CNN (see Fig 5) consisted of an input layer (28x28), 3 convolution layers, 3 max pooling layers, 2 dropout layers (0.25, 0.50), fully connected layers (128), and an output layer. By reducing the filter size, more detailed image features can be captured. In this study, this parameter was optimized to a 3x3 filter size. Since convolutional and pooling layers are composed of basic units, the number of convolutional layers and the size and number of each convolutional layer filter can affect the performance of the CNN.

A neural network structure as shown in Eq (3) was formed for the convolution operation in the CNN structure (W is the weight, x is the input, and b is the bias). In the middle step of the network, the ReLU function as in Eq (4) was used, and in the last step, the output was obtained

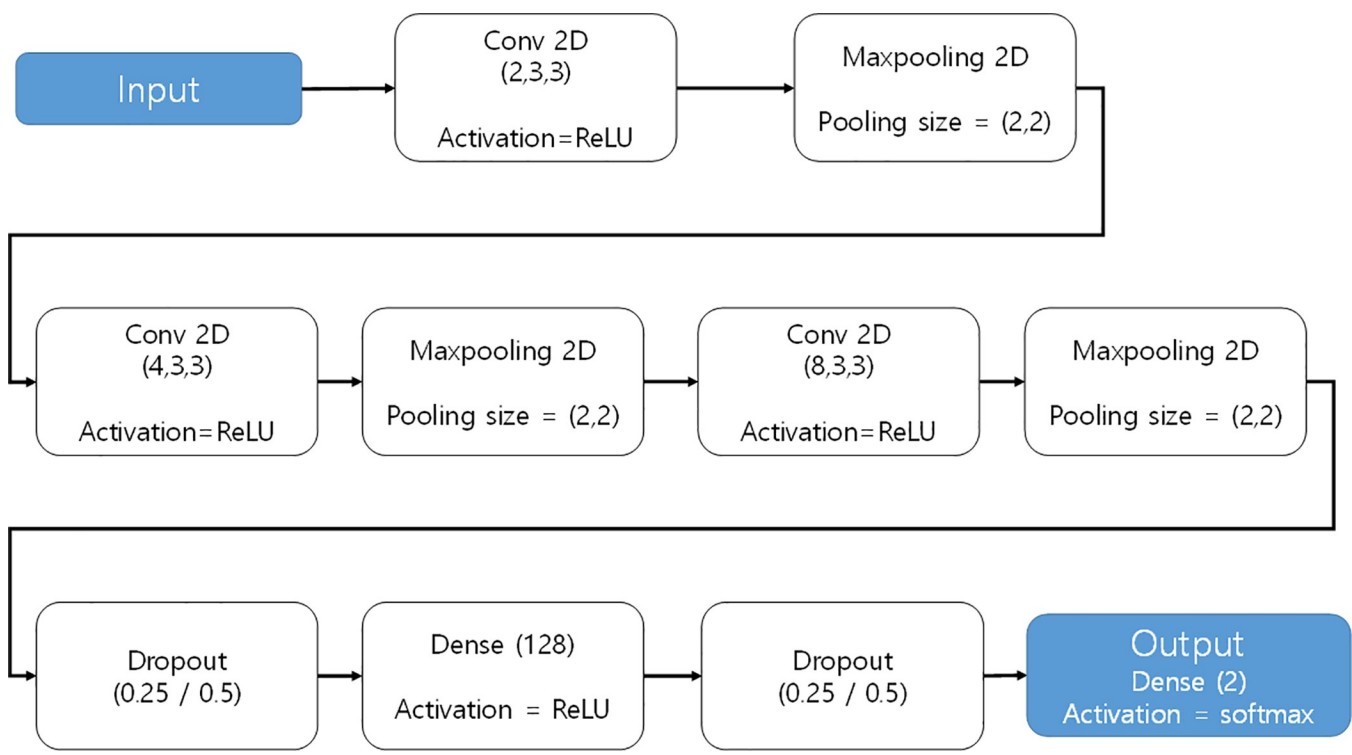

**Fig 5. Structure of the CNN used in the analysis.**

using softmax.

$$ei = \sum_j Wi, jXj + bi \tag{3}$$

$$relu\ f(x) = \begin{cases} 0, x < 0 \\ x, x \geq 0 \end{cases} \tag{4}$$

## 4.3 Evaluation methodology

Next, 10-fold cross validation was performed to determine the inference performance for each image with the image dataset secured for the test and the configured CNN model. At this point, evaluation metrics are needed to compare the results of our method with other methods. Accuracy is a common metric used for this purpose [2,11,12]. Accuracy was used in this paper because there is no imbalance issue in the data used in this study; thus, it is unnecessary to use the F-measure [7].

## 4.4 Experimental environment

To establish the experimental environment, we followed the method of Eapen [41]. We developed the model using the Python programming language (Python 3.7 in a Windows 64-bit system environment). In addition, development tools included PyCharm and Anaconda3; Keras was used to build the network model structure, and the TensorFlow framework was used at the bottom. The relevant parameters are outlined in Table 4.

We used RMSE as a measure of loss function and activation function. The image size was 28*28. Moreover, rather than using a standard CNN such as VGG and AlexNet, we tested the validity of our results on benchmark datasets.

Table 4. Experimental parameter settings.

| Hyperparameter | Values |
|---|---|
| Filter size | (3,3) |
| Size of max-pooling | (2,2) |
| Optimizer | Adam |
| Activation function | Activate function combinations between layers: {ReLU, Elu, Selu, Softsign, Softplus, RRelu, Gelu} Output: softmax |
| Learning rare | 0.001 |
| Epochs | 200 |
| Batch size | 200 |
| Padding | Same |
| Dropout rate | (0.5,0.25),(0.5,0.5) |
| Number of layers | 4 |

We used the following libraries for various functionalities:

- Keras [40], a popular open-source library that enables researchers and software engineers to define and train many deep learning models in a short amount of time. It is used for creating and training neural networks. It provides a simple interface to existing libraries like Tensorflow [42], which allows use of GPUs for faster training and prediction. With Keras, we can achieve fast experimentation, which is key to gaining insightful feedback and improving the accuracy of deep learning models for stock price prediction. Note that in Keras version 2.0, metrics such as recall, precision, and the F-measure have been removed to promote the use of accuracy as the main metric for CNNs that are trained on balanced datasets [40].

- Tensorflow [42], the backend for Keras. It facilitates the processing of low-level operations such as tensor products and convolutions. It is developed and maintained by Google.

- Scikit-learn [43] for performing the grid search of the model to find the best parameters using 10-fold cross-validation.

- Matplotlib [44] for plotting the graphs for the actual time series as well as predicted trends.

- Pandas [45] for reading values from csv files as DataFrames.

- Numpy [46] to perform matrix operations like flip and reshape and to create random matrices. Although there are various optimization algorithms such as Adam, AdaDelta, and RMSProp, Adam optimization was selected in our study following the protocol in a previous study [2]. The Adam optimization algorithm can be used instead of the classical SGD procedure to update network weights iteratively based on training data [47].

Hyperparameter optimization also affects the performance of CNNs [48,49]. A preliminary experiment was therefore conducted to select the hyperparameters for the CNNs that influence stock price prediction. In accordance with previous studies [48,50,51], among the possible convolution layers, pooling layers, dropouts, and filters, we focused on the number of filters and dropouts, examining the activation function for each layer as the hyperparameter to improve prediction. To determine whether the shape of dropouts and filters affects the accuracy of CNN inferences, the dropout variable was set to (0.25,0.5) and (0.5,0.5), and the number of 3-layer convolution kernels was set to (1,2,4), (2,4,8), (3,6,12), (4,8,16), (5,10,20), (6,12,24), (7,14,28), (8,16,32), (9,18,36), (10,20,40), (11,22,44), (12,24,48), (13,26,52), (14,28,56), (15,30,60), and (16,32,64). By designating the number of filters in this format, we

obtained 960 sample cases per image. The activation functions reviewed in the pretest to optimize the activation function in the CNN were ReLU, Elu, Selu, Softsign, Softplus, RRelu, and Gelu. As a result of experimenting with various combinations, we obtained the following sequence: [ELU, ELU, ELU, SELU]. We then applied 10-fold cross-validation for performance evaluation. Algorithm 1 summarizes the above experimental sequence.

```
Algorithm 1: Proposed Model Procedure
1: procedure StockPrediction()
2: Phase Data Preparation:
3: company = read(companyList, stockPrice)
4: trainingDataset = dataset.split(dates = 2015 – 2019)
5: testDataset = dataset.split(dates = 2015 – 2019)
6: Phase Labelling Data:
7: slopeRef[1..n] = calculateSlopeReferences(farFutureValue = 5,
nearFutureValue = 1)
8: calculate distribution of the class(Up, Down) to ind separation
values
9: firstSepPoint, secondSepPoint = find the separation values(slo-
peRef[1..n])
10: for(all dataset):
11: slopeCurrent = calculateEachImageSlope(farFutureValue = 5,
nearFutureValue = 1)
12: if(slopeCurrent > secondSepPoint):
13: label = 1 ("Up")
14: elif(slopeCurrent < = secondSepPoint):
15: label = 0 ("Down")
16: merge labels and images
17: Phase Chart Generation:
18: for(all chartNum):
19: for(all company):
20: for(all period):
21: chartProperty = getChartProperty(chartNum)
22: chart = generateChart(chartProperty, imagefile)
23: putImage(label, chart)
24: Phase Predicting Label:
25: model = CNN(epochs = 200, learningrate = 0.001, activation =
('Activation function combination', 'softmax'))
26: model.train(trainingDataset)
27: model.test(testDataset)
```

## V. Results

### 5.1 Differences in prediction accuracy according to image characteristics

In this study, stock price data from January 1, 2015 to December 31, 2019 were used in a CNN, and learning and prediction were performed by setting the closing price as 1 or 0 after 1 or 5 days. Table 5 shows the results of an ANOVA analysis comparing differences between factors according to the accuracy of the stock price prediction. The results of the analysis showed a significant difference in prediction accuracy for X1, X2, X4, X6, X7, X8, and X9. Putting together the results for overall accuracy and the ANOVA analysis, we see that when a stock price chart is used as a training dataset, the accuracy of the prediction can be increased by drawing a solid line, coloring the lower area, and using an image without axis marks. Therefore, Hypothesis 1 was partially supported.

### 5.2 Mediating role of filter

To test Hypothesis 2, which states that filter characteristics play a mediating role between image characteristics and stock price prediction accuracy, Chow's verification was performed by

**Table 5. Results of ANOVA analysis of the difference in accuracy between stock price predictions using image characteristics.**

| Variables | N | Accuracy | SD | F-value |
|---|---|---|---|---|
| X1 | Time series (308) | 0.639 | 0.048 | 6.744*** |
| | Frequency (112) | 0.626 | 0.043 | |
| X2 | Plot (112) | 0.643 | 0.050 | 5.263*** |
| | Histogram (112) | 0.630 | 0.041 | |
| | Bar plot (168) | 0.640 | 0.049 | |
| | Strip chart (28) | 0.608 | 0.038 | |
| X3 | Horizontal (238) | 0.635 | 0.0501 | 0.028 |
| | Vertical (182) | 0.636 | 0.0431 | |
| X4 | No (252) | 0.641 | 0.044 | 7.506*** |
| | Yes (168) | 0.628 | 0.050 | |
| X5 | After 1 day (210) | 0.636 | 0.048 | 0.019 |
| | After 5 days (210) | 0.635 | 0.046 | |
| X6 | Bar (308) | 0.633 | 0.046 | 3.388*** |
| | Line (112) | 0.643 | 0.050 | |
| X7 | No (238) | 0.629 | 0.049 | 10.295*** |
| | Yes (182) | 0.644 | 0.043 | |
| X8 | Monotone (252) | 0.629 | 0.048 | 12.077*** |
| | Color (168) | 0.645 | 0.044 | |
| X9 | No (252) | 0.640 | 0.042 | 5.632** |
| | Yes (168) | 0.629 | 0.053 | |

X1_Series, X2_Graph type, X3_Graph direction, X4_Appearance of axis, X5_Forecast date, X6_Bar/line, X7_Color, X8_Color tone, X9_Dotted/solid line.

classifying the filter variable into two groups: large (high filter) and small (low filter). The Chow's F(10,940) value was statistically significant at 5.223; therefore, Hypothesis 2 was supported. According to Table 6, which shows the results of the Chow verification, variables X1 and X5 had a positive effect on accuracy in the low-filter population, but a negative effect in the high-filter population. In addition, the X7 variable negatively affected accuracy in the low-filter population, but had a positive effect on the high-filter population. Both X4 and X6 positively affected accuracy, but the results show that the high filter had a greater effect than the low filter. Also, X2 negatively affected accuracy, but the high filter had a greater effect than the low filter. X1 and X2 had no effect in the low-filter condition, but had a negative effect in the high-filter condition. The X9 variable did not affect the accuracy, meaning that the filter size was not adjusted.

## 5.3 Mediating effect of dropout

To verify Hypothesis 3, which states that dropout acts as a mediator between image characteristics and stock price prediction accuracy, the dropout variable was divided into two groups: (0.25, 0.5) and (0.5, 0.5), and Chow verification was performed. The results show that Chow's F(10,940) = 2.3627 was statistically significant; therefore, Hypothesis 3 was supported (see Table 7). According to Table 7, variables X2, X3, and X6 did not affect either Dropout1 or Dropout2. It was found that X1, X4, X7 had more influence in Dropout1, X8 had a significant effect only on Dropout1, and X5 and X9 had a significant effect only on Dropout2.

## 5.4 Performance comparison

Using the results of hypothesis testing in this study, we compared the prediction performance of our method with those in other similar studies in terms of the chart image, filter, and

**Table 6. Results of Chow verification for filter.**

|  | Variables | B | SE | t | p |  |
|---|---|---|---|---|---|---|
| Low filter | (const) | 3.708 | 0.44 | 8.432 | 0 | 0.044 |
|  | X1 | -0.114 | 0.073 | -1.574 | 0.116 |  |
|  | X2 | -0.005 | 0.085 | -0.064 | 0.949 |  |
|  | X3 | 0.039 | 0.071 | 0.546 | 0.049** |  |
|  | X4 | 0.101 | 0.083 | 1.217 | 0.022** |  |
|  | X5 | 0.175 | 0.053 | 3.328 | 0.001*** |  |
|  | X6 | 0.064 | 0.196 | 0.326 | 0.074* |  |
|  | X7 | -0.032 | 0.14 | -0.231 | 0.008*** |  |
|  | X8 | -0.144 | 0.077 | -1.858 | 0.064* |  |
|  | X9 | -0.153 | 0.134 | -1.147 | 0.252 |  |
| High filter | (const) | 4.104 | 0.363 | 11.309 | 0 | 0.142 |
|  | X1 | -0.165 | 0.06 | -2.751 | 0.006*** |  |
|  | X2 | -0.136 | 0.071 | -1.932 | 0.044** |  |
|  | X3 | -0.136 | 0.059 | -2.32 | 0.021** |  |
|  | X4 | 0.514 | 0.069 | 7.476 | 0.000*** |  |
|  | X5 | -0.024 | 0.044 | -0.55 | 0.006*** |  |
|  | X6 | 0.278 | 0.162 | 1.713 | 0.087* |  |
|  | X7 | 0.402 | 0.116 | 3.475 | 0.001*** |  |
|  | X8 | -0.198 | 0.064 | -3.099 | 0.002*** |  |
|  | X9 | -0.072 | 0.11 | -0.649 | 0.517 |  |

Chow's $F(10,940)$: 5.223; p value: 0.0001.

X1_Series, X2_Graph type, X3_Graph direction, X4_Appearance of axis, X5_Forecast date, X6_Bar/line, X7_Color, X8_Color tone, X9_ Dotted/solid line.

dropout characteristics optimized for CNN-based stock price prediction. As shown in Table 8, the accuracy of the proposed method in this study was 64.3%, which was significantly higher than that of previous studies (52.1%–57.5%).

## VI. Discussion

### 6.1 Implications

As ways are developed to understand and analyze images through deep learning, research is being conducted to predict the ups and downs of stock prices using only the image of a stock price chart, not using an enormous amount of numerical information [3–5,10,30–33] that takes a lot of time and effort to collect and process [2,11,12]. In this study, we identify the specific characteristics of images that have a significant influence on the prediction accuracy of deep learning algorithms.

The greatest contribution of this study is its focus on the CNN algorithm to investigate the effects of the characteristics of stock price chart images on prediction performance. In fact, performing the CNN algorithm using image datasets or multimodal datasets has been actively attempted in various domains such as medicine [14]. In addition, prediction performance has been improved by selecting an appropriate preprocessing method according to the domain [52,53]. However, only a few studies have mentioned a preprocessing method that selects the optimal image characteristics in advance for stock price prediction [54].

In this study, we experimented with basic algorithms for CNNs under various conditions using a relatively large amount of image data (charts for training: 1,281,173, charts for testing: 142,352). The results revealed several significant variables: the type of chart, the use of dotted

Table 7. Results of Chow verification for dropout.

| | Variables | B | SE | t | p | |
|---|---|---|---|---|---|---|
| Dropout1 | (const) | 3.883 | 0.439 | 8.854 | 0 | 0.235 |
| | X1 | -0.149 | 0.072 | -2.051 | 0.041** | |
| | X2 | -0.06 | 0.085 | -0.703 | 0.482 | |
| | X3 | -0.069 | 0.071 | -0.981 | 0.327 | |
| | X4 | 0.322 | 0.083 | 3.878 | 0.000*** | |
| | X5 | 0.041 | 0.053 | 0.783 | 0.434 | |
| | X6 | 0.134 | 0.196 | 0.685 | 0.494 | |
| | X7 | 0.312 | 0.14 | 2.233 | 0.026** | |
| | X8 | -0.241 | 0.077 | -3.127 | 0.002*** | |
| | X9 | 0.06 | 0.133 | 0.451 | 0.652 | |
| Dropout2 | (const) | 3.929 | 0.377 | 10.413 | 0 | 0.286 |
| | X1 | -0.131 | 0.062 | -2.096 | 0.037** | |
| | X2 | -0.082 | 0.073 | -1.115 | 0.265 | |
| | X3 | -0.028 | 0.061 | -0.456 | 0.649 | |
| | X4 | 0.293 | 0.071 | 4.101 | 0.000*** | |
| | X5 | 0.11 | 0.045 | 2.439 | 0.015** | |
| | X6 | 0.207 | 0.168 | 1.231 | 0.219 | |
| | X7 | 0.058 | 0.12 | 0.478 | 0.0633* | |
| | X8 | -0.1 | 0.066 | -1.512 | 0.131 | |
| | X9 | -0.285 | 0.115 | -2.485 | 0.013** | |

Chow's F(10,940): 2.363764; p value: 0.0092.

X1_Series, X2_Graph type, X3_Graph direction, X4_Appearance of axis, X5_Forecast date, X6_Bar/line, X7_Color, X8_Color tone, X9_ Dotted/solid line.

or solid lines, indications of the area, and presence or absence of axis information. Based on the results of this analysis, we plan to develop an optimal method for determining the shape of the chart for providing to CNNs for the purpose of stock price prediction.

In addition, utilizing a CNN with selected chart images and specifying optimal filters and dropouts resulted in better prediction accuracy than in other, similar studies. This is the first study to show that stock price prediction is possible simply by observing the image characteristics of a chart, not relying on numerical information.

In the past, in addition to developing algorithms to improve stock price prediction, efforts have been made to improve feature selection on the premise that the quality of input data

Table 8. Comparative performance between previous and present results.

| Method | Data | Methods | Accuracy |
|---|---|---|---|
| Gunduz et al. [34] | Numeric | CNN | 56.0% |
| Nelson et al. [32] | Numeric | LSTM | 55.9% |
| Zhong & Enke [28] | Text | PCA | 57.5% |
| Gunduz et al. [34] | Numeric | CNN | 56.0% |
| Nelson et al. [32] | Numeric | LSTM | 55.9% |
| Zhong & Enke [28] | Text | PCA | 57.5% |
| Di Persio & Honchar [10] | Numeric | MLP | 52.1% |
| Di Persio & Honchar [10] | Numeric | LSTM | 52.2% |
| Di Persio & Honchar [10] | Numeric | CNN | 53.6% |
| Proposed method | Chart Image | Proposed | 64.3% |

characterization plays an important role in prediction performance [20]. This study supports the suggestion that considering the features of the image contributes to improving prediction performance.

## 6.2 Limitations

First, this study did not consider the patterns of stock price fluctuations. In future, patterns of stock price fluctuations will be meaningfully grouped for the purpose of pinpointing which image or CNN network characteristics contribute to improving prediction accuracy in which groups. Second, this study focused only on dropout and filter among the possible network characteristics of CNNs. However, there are various other points of comparison to study such as algorithm optimization, the activation function, and the number of layers. To focus on image characteristics in this study, we preoptimized some data in advance using Adam as the optimization algorithm, ReLU as the activation function, and including 5 layers (3 hidden layers). Future research will elucidate the role of these other parameters in prediction performance. Third, this study considered only CNNs among the possible deep learning algorithms. Although CNNs are the most commonly used image-based discrimination algorithms, and hence our choice was reasonable for this first study, it will be necessary to expand and study other image discrimination algorithms such as RNNs and LSTMs in future research. Last, only the CNN algorithm was used in our experiment, but in future work, we will also use algorithms such as BI-GRU and LSTM among RNNs. However, the main significance of this paper is its proof that there is a difference in the performance of deep learning algorithms according to image characteristics. We hope that tour results will inspire further research.

## 6.3 Conclusion

Stock price chart images are an important source of information for stock price forecasting. In the past, researchers had no choice but to use traditional classification algorithms that rely on numeric or textual data because there was no suitable preprocessing method to use image data. Recent development of deep learning algorithms such as CNNs has made it possible to discriminate using images and to make predictions using only stock price charts. This study reveals that there is an optimal fit between image characteristics and algorithm characteristics. The result (Hypothesis 1) suggests that we need to optimize the image characteristics for better stock price prediction. We identify a causal relationship between image characteristics and discriminant performance in order to improve the reliability and performance of image-based prediction. Moreover, the results of testing of Hypotheses 2 and 3 suggest that when selecting the characteristics of the stock chart, the filter parameter and dropout must also be changed according to the image characteristics.

Our method outperforms previous methods in terms of prediction accuracy. Through this study, image data may become widely used as an important resource for business intelligence. The results also suggest that there may be no need to prepare hybrid data (combining images and numerical data) for price prediction. Stock price prediction can be easy even in various formats of information systems (mobile apps, video-based SNS, image databases, etc.)–wherever chart images only are available. Although stock market prediction appears to be a random walk, a stock price expert looks at a company's stock price chart and predicts whether a company's stock price will rise or fall based on his knowledge and experience. We believe that the results of the performance test in this paper can be evaluated as showing the possibility that CNN can acquire knowledge and, to some extent, imitate the implicit knowledge of stock price prediction experts using charts. This paper assumes that people who predict stock prices using only charts without numerical data on stock prices have image-based forecasting

knowledge. In this study, the authors tried to augment their predictive ability through deep learning. It would be appreciated if you could understand the significance of this study in this way.

## Author Contributions

**Conceptualization:** Ohbyung Kwon.

**Data curation:** Guangxun Jin.

**Funding acquisition:** Ohbyung Kwon.

**Investigation:** Guangxun Jin.

**Project administration:** Ohbyung Kwon.

**Software:** Guangxun Jin.

**Supervision:** Ohbyung Kwon.

**Writing – original draft:** Guangxun Jin, Ohbyung Kwon.

**Writing – review & editing:** Ohbyung Kwon.

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
