## [Decision Letter · Decision Letter 0]

22 Dec 2020

PONE-D-20-29152

Impact of Chart Image Characteristics on Stock Price Prediction of a Convolutional Neural Network

PLOS ONE

Dear Dr. Kwon,

Thank you for submitting your manuscript to PLOS ONE. After careful consideration, we feel that it has merit but does not fully meet PLOS ONE’s publication criteria as it currently stands. Therefore, we invite you to submit a revised version of the manuscript that addresses the points raised during the review process.

The paper needs substantial improvements and explanations on multiple levels: the image data set generation and choice of features, use of technical indicators along image feature in tandem or in comparison, reformulation of the hypotheses, a more transparent model validation, a statistical comparison with state-of-the-art deep learning methods for the same data set, inclusion of more performance metrics.

We look forward to receiving your revised manuscript.

Kind regards,

Ruxandra Stoean

Academic Editor

PLOS ONE

Journal Requirements:

2. Please provide links to all data sources used in the Methods section.

4. Please upload a new copy of Figure 3 as the detail is not clear. Please follow the link for more information: https://blogs.plos.org/plos/2019/06/looking-good-tips-for-creating-your-plos-figures-graphics/

Reviewers' comments:

Reviewer's Responses to Questions

**Comments to the Author**

1. Is the manuscript technically sound, and do the data support the conclusions?

Reviewer #1: Partly

Reviewer #2: Yes

2. Has the statistical analysis been performed appropriately and rigorously? 

Reviewer #1: Yes

Reviewer #2: Yes

3. Have the authors made all data underlying the findings in their manuscript fully available?

Reviewer #1: No

Reviewer #2: Yes

4. Is the manuscript presented in an intelligible fashion and written in standard English?

Reviewer #1: Yes

Reviewer #2: Yes

5. Review Comments to the Author

Reviewer #1: In my view, the paper in its current form is in frontiers between major revision and direct rejection. The authors did not discuss the implementation of CNN in sufficient detail, lacking the parameters such as loss function, learning rates, and numbers of epochs, etc. There is no cohesion between the hypothesis and conclusion presented in the paper. However, the content of the paper shows a lot of work correctly oriented towards a goal. Hence, Major revision is suggested before it can be considered as competent for publication. The main concern that needs to be addressed are listed below:

1. I cannot see any novelty in this work. There are many publications in this regard except the dataset used.

2. Since dropout and filter-size are obvious parameters that affect CNN learning- so hypothesis 2 needs reformulation (hypothesis formulation adds no more value to this work).

3. There is no cohesion between the research hypothesis and conclusion.

4. Stock market prediction is a random walk and includes non-linear dynamics, authors are unable to address these issues in their experiments as well as conclusions.

5. Authors need to shed light on how chart-based stock prediction is feasible over technical indicator-based prediction.

6. The Paper lacks the details about image feature representation by CNN.

7. The information provided about image characteristic is ambiguous.

8. Model validation is weak. The author needs to specify the size of the increase dataset, decrease dataset, and Size of training, validation, and testing dataset explicitly.

9. Detail of CNN network training parameters is missing such as loss function.

10. Since the proposed CNN has very limited layers, the network overfitting might be the issue, the author needs to verify this with network generalization capability.

11. The comparison of the presented result with the previous result is not comprehensive. The reference cited in Table 7 (Namely Gunduz et.al -2017, Nelson et.al-2017) are missing in the reference list.

12. The validity of result presented in the paper need to be tested on benchmark datasets as author proposed their own CNN, instead of standard CNN such as VGG, and AlexNet, there are many CNN parameters still need to be optimized in the network such as numbers of layers, optimizers, activation functions, and normalizations.

13. The image dataset generation process needs to be discussed in detail.

14. The only accuracy is used as performance metrics which might be biased towards a single class either increases or decrease so the author needs to add a few more performance metrics.

15. There are many grammatical and syntactic errors inside. A native speaker can fix it.

Reviewer #2: PLOS ONE Review of the manuscript PONE-D-20-29152

The manuscript titled “Impact of Chart Image Characteristics on Stock Price Prediction of a Convolutional Neural Network” uses CNN on image features to forecast stock price. The research work presented in the manuscript is good by using image features for prediction. However, there are some weak points that should be addressed.

1. Scope of the manuscript is limited. The research work uses only image features for price prediction.

2. The manuscript uses only CNN on the image data for prediction. There are state-of-the-art methods that should also be used and compare with CNN.

3. The main idea of this research work is to use image features for prediction. But there is not enough information, how the images and there features are generated. So in section 4.1, please provide some more information how the chart images were created.

After the changes, the manuscript may be recommended for publication.

6. PLOS authors have the option to publish the peer review history of their article (what does this mean?). If published, this will include your full peer review and any attached files.

Reviewer #1: No

Reviewer #2: No

---

## [Author Response · Author response to Decision Letter 0]

23 Feb 2021

Comments Response

[Reviewr 2]

[Q1] I cannot see any novelty in this work. There are many publications in this regard except the dataset used. 

[A1] As you pointed out, the study of predicting stock prices with images has begun. Many example studies have been cited in the paper. However, there is still no research on ‘which’ characteristics of a chart image significantly affect the performance of deep learning-based stock price predictions. This is the novelty of this paper. This paper has been researched from this angle and its features are presented in the conclusion section.

[Q2] Since dropout and filter-size are obvious parameters that affect CNN learning- so hypothesis 2 needs reformulation (hypothesis formulation adds no more value to this work). 

[A2] As the reviewer mentioned, filter size and dropout are factors that influence the CNN model. In addition, in this paper, a hypothesis was created based on the idea that there will be a difference in the degree to which filter size and dropout affect each image feature. The result of Hypothesis 2 and 3 means that when selecting the characteristics of the stock chart, the filter parameter and dropout must also be changed according to the image characteristics. Therefore, the authors treated these parameters as mediating rather than independent variables. 

[Q3] There is no cohesion between the research hypothesis and conclusion. 

[A3] The authors modified the conclusion as advised, adding information about the hypotheses to the conclusion.

[Q4] Stock market prediction is a random walk and includes non-linear dynamics, authors are unable to address these issues in their experiments [A4] as well as conclusions. 

Although stock market prediction appears to be a random walk, a stock price expert looks at a company's stock price chart and predicts whether a company's stock price will rise or fall based on his knowledge and experience. The results of the performance test in this paper can be evaluated as showing the possibility that CNN can acquire knowledge to some extent imitate the implicit knowledge of stock price prediction experts using charts.

[Q5] This paper assumes that people who predict stock prices with only charts without numerical data on stock prices have image-based forecasting knowledge. 

[A5] In this study, the authors tried to augment their predictive ability through deep learning. It would be appreciated if you could understand the significance of this study in this way. These are newly added in conclusion.

[Q6] Authors need to shed light on how chart-based stock prediction is feasible over technical indicator-based prediction. 

[A6] Table 8 has been revised to show that using chart images results in more accurate stock price prediction than numeric and text-based prediction. 

The authors are considering a method of combining image-based prediction and technical indicator-based prediction for future research. Information to this effect has been provided in the discussion section of the revised version.

[Q7] The Paper lacks the details about image feature representation by CNN. 

[A7] The process of creating an image is as follows.

1. Daily data for 5 years (2015-2019) was collected on 789 companies included in the KOSPI Index calculation in the Dataguide program.

2. Images were composed of data collected on a monthly basis.

3. According to the image characteristics shown in Table 2, 30 images of various types were created for each month with RStudio software.

4. After the images were created, a label for each was automatically generated with the name of the image, which was then compared with the closing stock price 5 days later. Please see the following example. 

5. Finally, if the image name rate of return was greater than 0, 1 was automatically added to the end of the label; otherwise, 0 was automatically added.

This text has been added to section 4.1.

[Q8] The information provided about image characteristic is ambiguous. 

[A8] The information has been clarified in Table 2 and image characteristics have been explained better in the text. In the experiment for this paper, deep learning was performed by generating various chart images according to the characteristics listed in Table 2.

[Q9] Model validation is weak. The author needs to specify the size of the increase dataset, decrease dataset, and Size of training, validation, and testing dataset explicitly. 

[A9] In order to obtain a set of images for learning, a dataset was created with daily data for all companies listed on the KOSPI Index, and a stock price chart for each image characteristic was created. Total amount of data: 12 (months)*5 (years)*789 (number of companies)*30 (chart image types). Then, 90% of the training set and 10% of the test set were selected randomly so learning and inference could be performed.

This information has been added in the revised manuscript. 

[Q10] Detail of CNN network training parameters is missing such as loss function. 

[A10] We added the following information about parameters in the revised manuscript (p. 14).

Parameters: 

Loss function : RMSE

Activation function: {ReLU, softmax}

Image size = 28 * 28 

[Q11] Since the proposed CNN has very limited layers, the network overfitting might be the issue, the author needs to verify this with network generalization capability. 

[A11] The stock price images created for this study were relatively simple. In our experience, the more layers we set, the worse the performance, and the more parameters we included, the greater the number of calculations. Therefore, we concluded that there was no issue of overfitting.

[Q12] The comparison of the presented result with the previous result is not comprehensive. The reference cited in Table 7 (Namely Gunduz et.al -2017, Nelson et.al-2017) are missing in the reference list. 

[A12] Thank you. The reference is newly added.

[Q13] The validity of result presented in the paper need to be tested on benchmark datasets as author proposed their own CNN, instead of standard CNN such as VGG, and AlexNet, there are many CNN parameters still need to be optimized in the network such as numbers of layers, optimizers, activation functions, and normalizations. 

[A13] The authors optimized the network as follows:

Parameter: {Values}

Filter size: {3 x 3}

Size of max-pooling: {(2, 2)}

Optimizer: {Adam}

Activation function: {combination of activation function}

Epochs: {2000}

Batch size: {200}

Dropout rate: {0.5,0.25}

Number of layers: 3

This information has been added to the revised version of the paper (Table 4).

[Q14] The image dataset generation process needs to be discussed in detail. 

[A14] The process of creating an image is as follows.

1. Daily data for 5 years (2015-2019) was collected on 789 companies included in the KOSPI Index calculation in the Dataguide program.

2. Images were composed of data collected on a monthly basis.

3. According to the image characteristics shown in Table 2, 30 images of various types were created for each month with RStudio software.

4. After the images were created, a label for each was automatically generated with the name of the image, which was then compared with the closing price of the stock price 5 days later. Please see the following example. 

5. Finally, if the image name rate of return was greater than 0, 1 was automatically added to the end of the label; otherwise, 0 was automatically added.

This text has been added to section 4.1.

[Q15] There are many grammatical and syntactic errors inside. A native speaker can fix it. 

[A15] Thank you. The paper was edited by a native speaker who is familiar with technical papers.

[Reviewr 2]

[Q1] Scope of the manuscript is limited. The research work uses only image features for price prediction. 

[A1] The study of predicting stock prices with images has begun. Many example studies have been cited in the paper. However, there is still no research on how the characteristics of a chart affect the performance of deep learning-based stock price predictions. This is the novelty of this paper. 

The results (Table 8) suggest that using chart images is more useful for stock price prediction than numeric and text-based methods. 

Also, the authors are considering a method of combining image-based prediction and technical indicator-based prediction in future research. This information has been newly added in the discussion section.

[Q2] The manuscript uses only CNN on the image data for prediction. There are state-of-the-art methods that should also be used and compare with CNN. 

[A2] We agree with your statement. In the current study, only the CNN algorithm is used, but in future work, we will include algorithms such as BI-GRU and LSTM among RNNs. For the purposes of this paper, the main focus is differences in the performance of deep learning algorithms according to image characteristics. Our results inspire us to further research in this area.

This information has been added to the section discussing the limitations of this paper. 

[Q3] The main idea of this research work is to use image features for prediction. But there is not enough information, how the images and there features are generated. So in section 4.1, please provide some more information how the chart images were created. 

[A3] The process of creating an image is as follows.

1. Daily data for 5 years (2015-2019) was collected on 789 companies included in the KOSPI Index calculation in the Dataguide program.

2. Images were composed of data collected on a monthly basis.

3. According to the image characteristics shown in Table 2, 30 images of various types were created for each month with RStudio software.

4. After the images were created, a label for each was automatically generated with the name of the image, which was then compared with the closing stock price 5 days later. Please see the following example. 

5. Finally, if the image name rate of return was greater than 0, 1 was automatically added to the end of the label; otherwise, 0 was automatically added.

This text has been added in section 4.1.

---

## [Decision Letter · Decision Letter 1]

4 May 2021

PONE-D-20-29152R1

Impact of Chart Image Characteristics on Stock Price Prediction with a Convolutional Neural Network

PLOS ONE

Dear Dr. Kwon,

Thank you for submitting your manuscript to PLOS ONE. After careful consideration, we feel that it has merit but does not fully meet PLOS ONE’s publication criteria as it currently stands. Therefore, we invite you to submit a revised version of the manuscript that addresses the points raised during the review process.

ACADEMIC EDITOR:

Based on the comments received from the reviewers and my own observation, I recommend minor revisions for the manuscript.

We look forward to receiving your revised manuscript.

Kind regards,

Thippa Reddy Gadekallu

Academic Editor

PLOS ONE

Journal Requirements:

Reviewers' comments:

Reviewer's Responses to Questions

**Comments to the Author**

1. If the authors have adequately addressed your comments raised in a previous round of review and you feel that this manuscript is now acceptable for publication, you may indicate that here to bypass the “Comments to the Author” section, enter your conflict of interest statement in the “Confidential to Editor” section, and submit your "Accept" recommendation.

Reviewer #1: All comments have been addressed

Reviewer #3: (No Response)

2. Is the manuscript technically sound, and do the data support the conclusions?

Reviewer #1: Yes

Reviewer #3: Yes

3. Has the statistical analysis been performed appropriately and rigorously? 

Reviewer #1: Yes

Reviewer #3: Yes

4. Have the authors made all data underlying the findings in their manuscript fully available?

Reviewer #1: No

Reviewer #3: Yes

5. Is the manuscript presented in an intelligible fashion and written in standard English?

Reviewer #1: Yes

Reviewer #3: Yes

6. Review Comments to the Author

Reviewer #1: I appreciate that author attempted to address the reviewer's comments well in the revised manuscript and now the manuscript is much improved. However, the authors still didn't provide sufficient detail on model training and ablation study. It will be worthy for publication after such minor revision.

Reviewer #3: • The Wide ranges of applications need to be addressed in Introductions

• The objective of the research should be clearly defined in the last paragraph of the introduction section.

• Add the advantages of the proposed system in one quoted line for justifying the proposed approach in the Introduction section.

• The motivation for the present research would be clearer, by providing a more direct link between the importance of choosing your own method.

The authors can cite the following references

Analysis of dimensionality reduction techniques on big data

Deep neural networks to predict diabetic retinopathy

Antlion re-sampling based deep neural network model for classification of imbalanced multimodal stroke dataset

7. PLOS authors have the option to publish the peer review history of their article (what does this mean?). If published, this will include your full peer review and any attached files.

Reviewer #1: No

Reviewer #3: No

---

## [Author Response · Author response to Decision Letter 1]

14 May 2021

Reviewer 1

[Q1] The authors still didn't provide sufficient detail on model training and ablation study. It will be worthy for publication after such minor revision. 

[A1] Thank you for your comments. Details on model training have been added in Table 4 (activation function, learning rate, number of layers). 

Reviewer 2

[Q1] 

The objective of the research should be clearly defined in the last paragraph of the introduction section. • Add the advantages of the proposed system in one quoted line for justifying the proposed approach in the Introduction section. 

[A1] 

The objective of the research has been added in the last paragraph of the introduction.

The advantages of the proposed system have been outlined in the last paragraph of the introduction.

[Q2] 

The motivation for the present research would be clearer, by providing a more direct link between the importance of choosing your own method.

The authors can cite the following references

Analysis of dimensionality reduction techniques on big data

Deep neural networks to predict diabetic retinopathy

Antlion re-sampling based deep neural network model for classification of imbalanced multimodal stroke dataset • The motivation for our research has been described at the end of the introduction section.

[A2]

• The motivation for our research has been described at the end of the introduction section.

• The following sentences have been added in the revised version of the manuscript:

In fact, performing the CNN algorithm using image datasets or multimodal datasets has been actively attempted in various domains such as medicine (Reddy et al., 2020b). In addition, prediction performance has been improved by selecting an appropriate preprocessing method according to the domain (Gadekallu et al., 2020; Reddy et al., 2020a). However, only a few studies have mentioned a preprocessing method that selects the optimal image characteristics in advance for stock price prediction.

• Correspondingly, we also added the following references:

Reddy, G. T., Reddy, M. P. K., Lakshmanna, K., Kaluri, R., Rajput, D. S., Srivastava, G., & Baker, T. (2020a). Analysis of dimensionality reduction techniques on big data. IEEE Access, 8, 54776-54788.

Gadekallu, T. R., Khare, N., Bhattacharya, S., Singh, S., Maddikunta, P. K. R., & Srivastava, G. (2020). Deep neural networks to predict diabetic retinopathy. Journal Of Ambient Intelligence and Humanized Computing, 1-14.

Reddy, T., Bhattacharya, S., Maddikunta, P. K. R., Hakak, S., Khan, W. Z., Bashir, A. K., ... & Tariq, U. (2020b). Antlion re-sampling based deep neural network model for classification of imbalanced multimodal stroke dataset. Multimedia Tools and Applications, 1-25.

---

## [Editor Report · Decision Letter 2]

17 May 2021

PONE-D-20-29152R2

Impact of Chart Image Characteristics on Stock Price Prediction with a Convolutional Neural Network

PLOS ONE

Dear Dr. Kwon,

Thank you for submitting your manuscript to PLOS ONE. After careful consideration, we feel that it has merit but does not fully meet PLOS ONE’s publication criteria as it currently stands. Therefore, we invite you to submit a revised version of the manuscript that addresses the points raised during the review process.

ACADEMIC EDITOR:

I guess the authors submitted the revision in a hurry. They claimed that they have addressed all the comments of teh reviewers in the response sheet but in the manuscript they are not reflected. For instance in teh response sheet the authors claimed that:

• Correspondingly, we also added the following references: Reddy, G. T., Reddy, M. P. K., Lakshmanna, K., Kaluri, R., Rajput, D. S., Srivastava, G., & Baker, T. (2020a). Analysis of dimensionality reduction techniques on big data. IEEE Access, 8, 54776-54788. Gadekallu, T. R., Khare, N., Bhattacharya, S., Singh, S., Maddikunta, P. K. R., & Srivastava, G. (2020). Deep neural networks to predict diabetic retinopathy. Journal Of Ambient Intelligence and Humanized Computing, 1-14. Reddy, T., Bhattacharya, S., Maddikunta, P. K. R., Hakak, S., Khan, W. Z., Bashir, A. K., ... & Tariq, U. (2020b). Antlion re-sampling based deep neural network model for classification of imbalanced multimodal stroke dataset. Multimedia Tools and Applications, 1-25

But this is not reflected in the paper. I recommend the authors to check all their manuscript carefully, address  all the comments and then submit the revised manuscript

We look forward to receiving your revised manuscript.

Kind regards,

Thippa Reddy Gadekallu

Academic Editor

PLOS ONE
---

## [Author Response · Author response to Decision Letter 2]

28 May 2021

[Q]

They claimed that they have addressed all the comments of the reviewers in the response sheet but in the manuscript they are not reflected. For instance in teh response sheet the authors claimed that:

Correspondingly, we also added the following references: Reddy, G. T., Reddy, M. P. K., Lakshmanna, K., Kaluri, R., Rajput, D. S., Srivastava, G., & Baker, T. (2020a). Analysis of dimensionality reduction techniques on big data. IEEE Access, 8, 54776-54788. Gadekallu, T. R., Khare, N., Bhattacharya, S., Singh, S., Maddikunta, P. K. R., & Srivastava, G. (2020). Deep neural networks to predict diabetic retinopathy. Journal Of Ambient Intelligence and Humanized Computing, 1-14. Reddy, T., Bhattacharya, S., Maddikunta, P. K. R., Hakak, S., Khan, W. Z., Bashir, A. K., ... & Tariq, U. (2020b). Antlion re-sampling based deep neural network model for classification of imbalanced multimodal stroke dataset. Multimedia Tools and Applications, 1-25

But this is not reflected in the paper. 

[A]

Thank you for your careful review. We added missing references and modified the bibliography form to fit PLOS-ONE. All other comments are already reflected from the R2 version of the paper.

---

## [Editor Report · Decision Letter 3]

1 Jun 2021

Impact of Chart Image Characteristics on Stock Price Prediction with a Convolutional Neural Network

PONE-D-20-29152R3

Dear Dr. Kwon,

We’re pleased to inform you that your manuscript has been judged scientifically suitable for publication and will be formally accepted for publication once it meets all outstanding technical requirements.

Kind regards,

Thippa Reddy Gadekallu

Academic Editor

PLOS ONE
---

## [Editor Report · Acceptance letter]

3 Jun 2021

PONE-D-20-29152R3 

Impact of Chart Image Characteristics on Stock Price Prediction with a Convolutional Neural Network 

Dear Dr. Kwon:

I'm pleased to inform you that your manuscript has been deemed suitable for publication in PLOS ONE. Congratulations! Your manuscript is now with our production department. 

Kind regards, 

on behalf of

Dr. Thippa Reddy Gadekallu 

Academic Editor

PLOS ONE